**Data Availability Statement:** All data were accessed from public repositories. Brazilian Institute of Geography and Statistics (IBGE) (https://www.ibge.gov.br/), the Department of

# First year of COVID-19 in Brazil: Factors associated with the spread of COVID-19 in small and large cities

**Alexandre Augusto de Paula da Silva**[1,2☯]*, **Rodrigo Siqueira Reis**[2,3☯], **Franciele Iachecen**[4☯], **Fábio Duarte**[5☯], **Cristina Pellegrino Baena**[1☯], **Adriano Akira Ferreira Hino**[1,2,4☯]

1 School of Medicine, Graduate Program in Health Sciences, Pontifícia Universidade Católica do Paraná, Curitiba, Paraná, Brazil, 2 Research Group of Physical Activity and Quality of Life, Pontifícia Universidade Católica do Paraná, Curitiba, Paraná, Brazil, 3 People Health and Place Unit, Prevention Research Center, Brown School. Washington University in Saint Louis, St. Louis, Missouri, United States of America, 4 Graduate Program in Health Technology, Polytechnic School, Pontifícia Universidade Católica do Paraná, Curitiba, Paraná, Brazil, 5 Department of Urban Studies and Planning, Massachusetts Institute of Technology, Cambridge, Massachusetts, United States of America

☯ These authors contributed equally to this work.
* alexandre.augustosilva@outlook.com

## Abstract

### Aim

To test the association between sociodemographic and social characteristics with COVID-19 cases and deaths in small and large Brazilian cities.

### Methods

This ecological study included COVID-19 data available in State Health Secretaries (managed by brasil.io API) and three national databases (IBGE, DATASUS and Embrapa). Temporal spread of COVID-19 in Brazil during the first year considered as outcome: a) days until 1st case in each city since 1st in the country; b) days until 1,000 cases/100,000 inhabitants since 1st case in each city; c) days until 1st death until 50 deaths/100,000 inhabitants. Covariates included geographic region, city social and environmental characteristics, housing conditions, job characteristics, socioeconomic and inequalities characteristics, and health services and coverage. The analysis were stratified by city size into small (<100,000 inhabitants) and large cities (≥100,00 inhabitants). Multiple linear regressions were performed to test associations of all covariates to adjust to potential confounders.

### Results

In small cities, the first cases were reported after 82.2 days and 1,000 cases/100,000 were reported after 117.8 days, whereas in large cities these milestones were reported after 32.1 and 127.7 days, respectively. For first death, small and large cities took 121.6 and 36.0 days, respectively. However, small cities were associated with more vulnerability factors to first case arrival in 1,000 cases/100,000 inhabitants, first death and 50 deaths/100,000

Informatics of Brazilian Public Health System –
DATASUS, Ministry of Health (https://datasus.
saude.gov.br/), the Brazilian Agricultural Research
Corporation (Embrapa) (https://www.embrapa.br/)
and from Brazil.io (https://brasil.io/home/). Full
paper dataset can be accesss here - doi.org/10.
6084/m9.figshare.24999911

**Funding:** AAPS: received PhD scholarship from
Coordenação de Aperfeiçoamento de Pessoal de
Nível Superior – Brasil (CAPES) – Finance Code
001 RSR: was supported by the Centene Charitable
Foundation for travel support by the funder and no
salary or any compensation was provided by this
agreement. The funders had no role in study
design, data collection and analysis, decision to
publish, or preparation of the manuscript.

**Competing interests:** RSR: was supported by the
Centene Charitable Foundation, founded by the
Centene Corporation, for travel support by the
funder and no salary or any compensation was
provided by this agreement. This does not alter our
adherence to PLOS ONE policies on sharing data
and materials. There are no patents, products in
development or marketed products associated with
this research to declare.

inhabitants. North and Northeast regions positively associated with faster COVID-19 incidence, whereas South and Southeast were least.

## Conclusion

Social and built environment characteristics and inequalities were associated with COVID-19 cases spread and mortality incidence in Brazilian cities.

## Background

Since late 2019 when the SARS-CoV-2 virus emerged in Wuhan province (China), the spread rapidly reached global scale and COVID-19 disease has impacted nearly all countries in world [1]. The virus has high capacity of transmission by aerosol droplets, contaminated surfaces or close contact with infected individuals [2]. Worldwide, more than 545,000,000 cases have been identified and nearly six million deaths, as June 2022 [3].

In March 2020's, the World Health Organization declared a global COVID-19 pandemic and recommended to governments around the world to adopt local restrictions to mitigate or suppress the virus transmission to slow community infection, and, consequently, deaths [4]. Social distancing, quarantine, controlled availability of services (e.g. public transit), reduced urban mobility, face masking and frequent hygiene behaviors were some of actions proposed and often adopted to help reducing community transmission [5]. These actions also aimed at providing clinical care for the vulnerable and at risk groups and those providing "essential" services (e.g., health care, food supply chain and public transit workers, older adults and people with morbidities) [6].

In Latin America the first COVID-19 positive case was reported in February 26[th], 2020 in São Paulo, Brazil [7, 8]. By the end of March 2021, Brazil was among the countries with the highest absolute number of cases and deaths (>12,200,000 cases and >300,000 deaths), and lethality (2.5%) [3]. In addition, underreporting among asymptomatic individuals was widely documented [9, 10]. The rapid spread of COVID-19 across the country was a concern to many public health researchers and officers [11, 12]. The absence of mass testing, poor coordination between health agencies, and inconsistent implementation of mitigation strategies have been suggested as probable causes of the quick spread [12].

Social determinants of health and social, health and economic inequalities including poor housing conditions, household overcrowding, poverty and low income, lack of health insurance, low levels of schooling and higher levels of urbanization [13] have also been identified as contributors to the COVID-19 spread and mortality [14–18]. In addition, racial and ethnic disparities have also been documented as associated with COVID-19 morbidity and mortality [13, 14]. In Brazil, poor access to hospital care, including Intensive Care Units (ICU), have been associated with both COVID-19 illness and deaths [18], suggesting major health and social inequities have played a role on the burden of the disease in the country [19].

Previous studies have examined social and health disparities associated with COVID-19 morbidity and mortality at State level or in few cases in large metropolitan areas [15] and at one point in time only [15, 16]. Therefore, there is little understanding about social and environmental factors associated with COVID-19 spread in the whole country. In fact, less than one out of ten Brazilians cities have more than 100,000 inhabitants [20]. However, city level social and environmental inequalities and their associations with disease changes over time in Brazil have not yet been examined.

Although within cities inequalities are observed in larger cities variations are higher between small and large cities. Smaller cities are often operating with limited funding to support health and transportation systems [21] whereas larger cities have better health care, transportation, and job access and opportunities. The aim of this study is to analyze the association between social, health and environmental characteristics, and COVID-19 morbidity and mortality in the first year of the pandemic in Brazil in small and large cities.

## Methods

This was an ecological longitudinal study [22] using city level open access data from 26 States and the Federal District and from the Brazilian Institute of Geography and Statistics (IBGE) [20], the Department of Informatics of Brazilian Public Health System–DATASUS, Ministry of Health [23], the Brazilian Agricultural Research Corporation (Embrapa) [24] and from Brazil.io [25]. Data from all 5,570 cities in Brazil were included in the analysis.

COVID-19 data included cases and deaths reported between February 26th, 2020 and February 4th, 2021 [25]. COVID-19 vaccinations started in January 2021. The following outcomes were computed: a) days between the first case in Brazil until the first case in the city; b) days between the first case in the city until the day when 1,000 cases were reported; and c) days between the first death in city until the day when 50 deaths inhabitants were reported. Descriptive analyses were performed on the following: proportion of cities reaching 1,000 cases; number of cases at three, six, nine and 12 months after first case; cities reporting at least one COVID-19 related death; number of COVID-19 related deaths at three, six, nine and 12 months after first death in the country. All incidence data is adjusted for 100,000 inhabitants. These outcomes were considered based on the chain of transmission by infectious agents and relevant incidences [26].

The following covariates were included: a) geographic region where the city is located (Midwest, North, Northeast, Southeast and South), metropolitan city (no/yes) and urban or rural; b) social and environmental city characteristics [total area ($Km^2$), urban area ($Km^2$), population size (inhabitants), population living within urban area (inhabitants), population older than 60 years (%), indigenous population (%), black population (%), illiterate older than 25 years (%) and city in extreme poverty (no/yes)]; c) housing conditions [household with density >2 per dormitory (%), household with garbage collection (%), household connected to the water supply system (%) and household connected to the sewer system (%)]; d) job characteristics [commerce (%) and informal workers (%)]; e) socioeconomic and inequalities characteristics [GINI index; income per capita; poor or extremely poor (%) and households in informal urban settlements (%)]; f) health services access and coverage [number of National Public Health System (SUS) physicians per inhabitants (100,000 inhabitants), number of SUS nurses per inhabitants (100,000 inhabitants), number of intensive care units or ICU per inhabitants (100,000 inhabitants). All health services access and coverage variables were standardized using z-scores, combined into one single variable categorized into tertiles. All variables were selected in according to literature review[13–15, 18].

### Statistical analysis

All data linkage was processed using the software QilkSense Desktop and analyzed using SPSS 23.0. Descriptive statistics (e.g., frequency distribution, mean and standard deviation) were presented, bivariate correlation matrix (S1 Table) and Variance Inflation Factor (VIF) was used to test multicollinearity between independent variables (S2 Table). Linear regressions were performed to analyze associations between independent variables and main outcomes in

according to city population size ($<$100,000 inhabitants and $\geq$100,000 inhabitants). Final models were adjusted for all independent variables using enter method.

## Results

Brazil has 5,570 cites with an estimated total population of 210 million inhabitants in 2019. Cities are mostly located in the Southeast region (41.8%), followed by Northeast (27.5%), South (14.3%), North (8.7%) and Midwest (7.7%), metropolitan cities (25.7%), urban (92.1%) and have predominantly $\leq$100,000 inhabitants (94.2%). Overall, cities with 100,000 inhabitants or more have higher population density, crowding ($>$2 persons per dormitory), access to sanitation (garbage collection, water, and sewer services), health services and infrastructure (physicians, nurses, and ICU units), jobs and income. Cities with smaller population size have higher proportion of black and indigenous residents, and illiteracy and poverty rates (Table 1).

In cities of 100,000 inhabitants or more, COVID-19 cases were reported nearly one-month (32.1±10.4 days) after the first case was reported in the country, whereas in smaller cities the first cases were reported after 82.2 (±32.4) days. Nonetheless, one year after COVID-19 was first reported in Brazil nearly all cities reported at least one case. In larger cities (100,000 inhabitants or more) the first COVID-19 related death was reported one month after the first case (36.0±22.4 days), whereas in smaller cities the first death was reported nearly months after (121.6±69.3 days). Finally, similar pattern was observed number of days to reach 50 deaths/ 100,000 inhabitants), with larger cities reporting earlier those numbers earlier (109.6 days) than smaller cities (128.4 days) (Table 1) (Fig 1). Additional data are available about the advance of COVID-19 every three, six, nine and twelve months (S2 Fig).

Results showed, in days, the association of first (less days) or late reported (more days) of the outcomes analyzed. Overall, in smaller cities ($<$100,000 inhabitants) the first cases were first reported, in North and Northeast regions (β-26.293 (2.077) days and β-19.052 (2.217) days, respectively), rural, and in the cities with higher urbanization, and crowding, and in those with greater access to sanitation, commercial related jobs and health. In addition, cities cases were also late reported in cities with lower proportion of black residents and older adults. In larger cities ($\geq$100,000 inhabitants), COVID-19 cases were first reported in those cities from rural areas, with lower proportion of black residents, lower income, crowding and late reported in cities with high illiteracy. (p$<$0.05) (Table 2).

In small cities ($<$100,00 inhabitants) the number of days to reach 1,000 cases/100,000 inhabitants were reported first in the Northeast region, rural, greater illiteracy, crowding, water supply, income per capita, informal settlements, and health coverage (p$<$0.05). The 1,000 cases/100,000 inhabitants were reached later in Southeast and South regions, metropolitan cities, and in cities with higher proportion of older adults, and in cities with greater proportion of those living in extreme poverty and inequality (GINI), commerce workers, and with access to garbage services. In larger cities, the 1,000 cases/100,000 inhabitants were first reported in the Northeast region, in cities with higher illiteracy, per capita income, otherwise, reached after in cities in the South region, with greater proportion of older adults and black residents (Table 3).

The first 50 deaths/100,000 inhabitants were reported earlier in the small cities ($<$100,000 inhabitants) with high proportion of older adults, and illiterate population, with water supply, informal workers, income, and informal urban settlements and reached after in cities located in the Southeast and South regions, and with higher proportion of indigenous population, and higher access to sanitation, commerce workers, and inequalities (GINI). Finally, in larger cities 50 deaths/100,000 inhabitants were reached earlier in metropolitan cities and higher illiteracy,

**Table 1. Descriptive characteristics of COVID-19 and cities in according to population size (n = 5.570).**

| | ≤100.000 (n = 5,246) | | >100.000 (n = 324) | |
|---|---|---|---|---|
| Total (n, %) | 5,246 | 94.2 | 324 | 5.8 |
| **COVID-19** | | | | |
| **Days until 1st case (mean, sd)** | 82.2 | 32.4 | 32.1 | 10.4 |
| **1.000 cases/100,000** | | | | |
| Not reached (n, %) | 281 | 5.4 | 321 | 0.9 |
| Reached (n, %) | 4,965 | 94.6 | 3 | 99.1 |
| **Days until 1,000 cases/100,000 (mean, sd)** | 117.8 | 59,2 | 127.7 | 41.3 |
| **At least 1 death** | | | | |
| No (n, %) | 388 | 7.4 | 0 | 0.0 |
| Yes (n, %) | 4,858 | 92.6 | 324 | 100.0 |
| **Days until 1st death (mean, sd)** | 121.6 | 69.3 | 36.0 | 22.4 |
| **50 deaths/100,000** | | | | |
| No deaths (n, %) | 388 | 7.4 | 0 | 0.0 |
| Not reached (n, %) | 1,818 | 34.7 | 16 | 4.9 |
| Reached (n, %) | 3,040 | 57.9 | 308 | 95.1 |
| **Days until 50 deaths/100,000 (mean, sd)** | 109.6 | 68.7 | 128.4 | 57.1 |
| **GEOGRAPHIC REGION** | | | | |
| Midwest (n, %) | 443 | 8.4 | 24 | 7.4 |
| North (n, %) | 1,731 | 33.0 | 63 | 19.4 |
| Northeast (n, %) | 420 | 8.0 | 30 | 9.3 |
| Southeast (n, %) | 1,514 | 28.9 | 154 | 47.5 |
| South (n, %) | 1,138 | 21.7 | 53 | 16.4 |
| Metropolitan area | | | | |
| No (n, %) | 4,017 | 76.6 | 121 | 37.3 |
| Yes (n, %) | 1,229 | 23.4 | 203 | 62.7 |
| Urban or Rural | | | | |
| Urban (n, %) | 4,805 | 91.7 | 321 | 99.1 |
| Rural (n, %) | 436 | 8.3 | 3 | 0.9 |
| **SOCIAL AND ENVIRONMENTAL CHARACTERISTICS** | | | | |
| Total area (km$^2$) (mean, sd) | 1,452.44 | 5,030.60 | 2,709.49 | 11,414.97 |
| Urban area (km$^2$) (mean, sd) | 4,421.49 | 7,516.76 | 74,490.48 | 105,631.75 |
| Population size (mean, sd) | 17,049.69 | 17,670.71 | 372,544.60 | 849,188.63 |
| Population living in urban area (mean, sd) | 10,780.18 | 13,514.72 | 322,245.04 | 777,923.64 |
| Populational density (inh./Km$^2$) (mean, sd) | 48.37 | 115.24 | 1,276.09 | 2,269.38 |
| Population older than 60 years (%) (mean, sd) | 12.23 | 3.25 | 9.81 | 2.66 |
| Indigenous population (%) (mean, sd) | 0.75 | 4.47 | 0.23 | 0.48 |
| Black population (%) (mean, sd) | 51.53 | 23.75 | 48.35 | 20.33 |
| Illiterate older than 25 years (%) (mean, sd) | 21.25 | 12.67 | 8.47 | 6.14 |
| City in extreme poverty | | | | |
| No (n, %) | 3,669 | 69.9 | 319 | 98.5 |
| Yes (n, %) | 1,577 | 30.1 | 5 | 1.5 |
| **HOUSING CONDITIONS** | | | | |
| Household with density >2 per dormitory (%) (mean, sd) | 24.95 | 13.08 | 27.86 | 11.24 |
| Household with garbage collection (%) (mean, sd) | 93.83 | 11.30 | 97.58 | 3.75 |
| Household connected to the water supply (%) (mean, sd) | 68.54 | 19.10 | 86.39 | 16.69 |
| Household connected to the sewer system (%) (mean, sd) | 28.25 | 30.69 | 59.25 | 30.85 |
| **JOB CHARACTERISTICS** | | | | |

*(Continued)*

**Table 1.** (Continued)

| | ≤100.000 (n = 5,246) | | >100.000 (n = 324) | |
|---|---|---|---|---|
| Commerce (%) (mean, sd) | 10.16 | 4.14 | 17.30 | 2.86 |
| Informal workers (%) (mean, sd) | 25.69 | 9.85 | 17.64 | 5.93 |
| **SOCIOECONOMIC AND INEQUALITIES CHARACTERISTICS** | | | | |
| GINI Index (2010) (mean, sd) | 0.501 | 0.066 | 0.525 | 0.055 |
| Income per capita (mean, sd) | 474.62 | 225.19 | 801.31 | 309.56 |
| Poor or extremely poor (%) (mean, sd) | 32.52 | 17.35 | 16.57 | 8.72 |
| Informal urban settlements (%) (mean, sd) | 0.60 | 3.2 | 8.0 | 11.2 |
| **HEALTH SERVICES ACCESS AND COVERAGE** | | | | |
| SUS Physicians/100,000 (mean, sd) | 68.06 | 66.48 | 153.02 | 98.33 |
| SUS Nurses/100,000 (mean, sd) | 90.30 | 44.36 | 111.88 | 55.08 |
| ICU/100,000 (mean, sd) | 1.30 | 7.67 | 17.72 | 18.96 |

crowding and informal urban settlements, while reached later in higher urban population (Table 4).

## Discussion

We examined the association between social, health, environmental characteristics, and COVID-19 morbidity and mortality in the first year of the pandemic in Brazil in small and large cities. We found that COVID-19 cases and deaths reached larger cities (32.1 and 36.0 days) faster than smaller ones (82.2 and 121.6 days). Overall, high income and schooling levels were associated with the speed in which COVID-19 reached and spread within cities regardless population size (S2 Fig). In small cities (<100,000 inhabitants) geographic location (Northeast region), metropolitan, rural, higher levels of population density and elderly population, crowding, access to clean water, health service coverage, and greater proportion of informal urban settlements were associated with COVID-19 reach and spread. COVID-19 related deaths increased faster in cities from metropolitan areas, with higher illiteracy levels and greater proportion of informal urban settlements regardless population size. In addition, in small cities (<100,000 inhabitants) these factors included geographic location (Northeast region), higher proportion of elderly, informal workers population, and lower proportion of indigenous and commerce related jobs, household with sewer system and GINI index. Finally, in greater cites (>100,000 inhabitants) higher levels of crowding and income were associated with COVID-19 related deaths speed of occurrence.

At the early stages of the COVID-19 pandemic, Brazilian state capitals and greater metropolitan areas were the main port of entrance for international travelers and where the first cases were registered [27–29]. In fact, our findings mirror those observed at the beginning of the pandemic when first cases were registered in the North and Northeast regions, and in cities with higher urbanization and crowding, access to sanitation, commercial related jobs, and health [16, 30]. This pattern of spreading has not changed overtime and it suggests that proximity of larger metropolitan, and a combination of urban and social environmental characteristics could lead to a faster spread of the disease. Similarly, we've found that the most vulnerable populations were first reached by the COVID-19 pandemic in Brazil, including indigenous and elderly and those living in areas where environmental conditions are worsened. Similar findings have been documented in Brazil and elsewhere, confirming that worsened social and environmental conditions could greatly contribute to the speed of the disease spread [13, 14, 16, 31–34]. However, it is important to consider the unique context of some

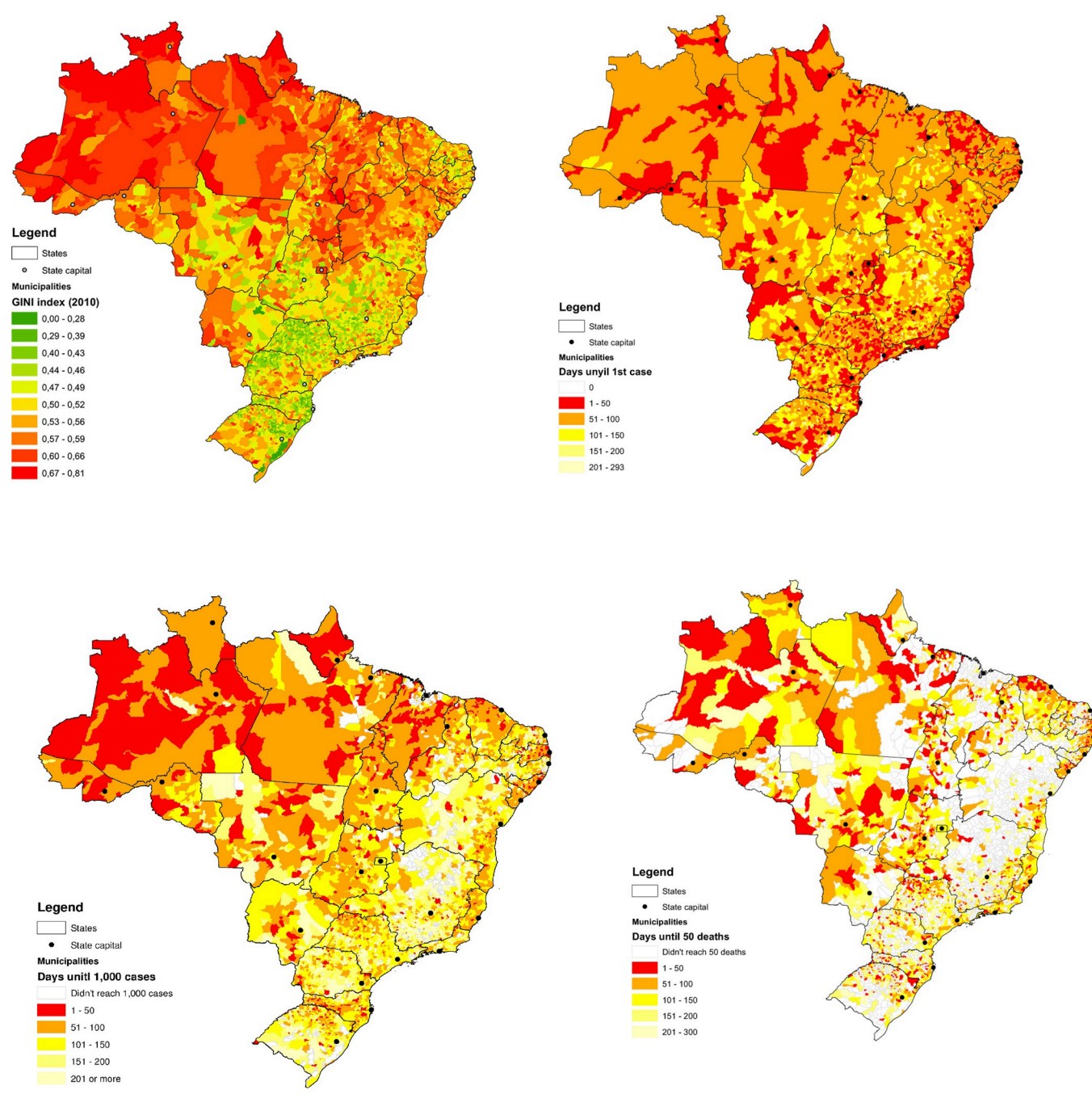

**Fig 1.**

regions, such as small southern cities with meatpacking plants. Since the food production sector is deemed an essential activity, industries like meatpacking plants were not required to adhere to most of the lockdown measures imposed during the pandemic. This led to increased exposure for its workers and their families, ultimately resulting in a local surge in COVID-19 cases [35].

Similarly, COVID-19 related deaths impacted larger cities greatly than small cities. This is somehow expected as it follows the early arrival of the first cases. In addition, arrival of

**Table 2. Factors associated to number of days since first case of COVID-19 in Brazil until first case in city (n = 5,118).**

| | | <100,000 | | | | ≥100,000 | | | |
|---|---|---|---|---|---|---|---|---|---|
| | | β (Std. error) | Std. Beta | t | p | β (Std. error) | Std. Beta | t | p |
| **GEOGRAPHIC REGION** | | | | | | | | | |
| Region (ref = Midwest) | North | -26.293 (2.077) | -0.382 | -12.657 | **<0.001** | -2.950 (2.741) | -0.112 | -1.076 | 0.283 |
| | Northeast | -19.052 (2.217) | -0.157 | -8.593 | **<0.001** | 0.593 (2.970) | 0.016 | 0.200 | 0.842 |
| | Southeast | -2.533 (1.939) | -0.036 | -1.306 | 0.192 | 3.181 (2.487) | 0.152 | 1.279 | 0.202 |
| | South | -1.957 (1.958) | -0.025 | -1.000 | 0.317 | -5.048 (2.715) | -0.179 | -1.860 | 0.064 |
| Metropolitan (ref = no) | | 8.900 (1.552) | 0.075 | 5.736 | **<0.001** | 10.186 (5.180) | 0.093 | 1.966 | 0.050 |
| Urban or rural (ref = urban) | | -3.928 (0.948) | -0.051 | -4.143 | **<0.001** | -2.744 (1.307) | -0.127 | -2.100 | **0.037** |
| **SOCIAL AND ENVIRONMENTAL CHARACTERISTICS** | | | | | | | | | |
| Urban population** (ref = low)[a] | Medium | 1.104 (1.022) | 0,025 | 1.597 | 0.110 | -4.602 (4.015) | -0.093 | -1.146 | 0.253 |
| | High | -0.182 (1.393) | 0,073 | 4.033 | **<0.001** | -5.306 (4.021) | -0.128 | -1.320 | 0.188 |
| Population older than 60 years** (ref = low)[a] | Medium | 1.716 (1.074) | -0,057 | -4.570 | **<0.001** | 2.147 (1.317) | 0.096 | 1.630 | 0.104 |
| | High | 4.950 (1.227) | -0,010 | -0.513 | 0.608 | 1.912 (2.103) | 0.050 | 0.909 | 0.364 |
| Indigenous population[a] | | -2.262 (0.495) | 0,042 | 1.781 | 0.075 | -1.590 (1.142) | -0.075 | -1.392 | 0.165 |
| Black population** (ref = low)[a] | Medium | -0.666 (1.300) | 0,049 | 2.666 | **0.008** | -6.222 (1.593) | -0.281 | -3.907 | **<0.001** |
| | High | 2.863 (1.607) | 0,019 | 0.638 | 0.524 | -4.261 (2.175) | -0.192 | -1.959 | 0.051 |
| Illiterate older than 25 years** (ref = low)[a] | Medium | 3.369 (1.264) | -0,017 | -1.232 | 0.218 | 4.096 (2.032) | 0.144 | 2.015 | **0.045** |
| | High | 1.283 (2.012) | 0,016 | 1.080 | 0.280 | 13.470 (3.810) | 0.212 | 3.536 | **<0.001** |
| City in extreme poverty (ref = no) | | -1.179 (0.957) | -0,003 | -0.131 | 0.896 | 2.836 (4.245) | 0.033 | 0.668 | 0.505 |
| **HOUSING CONDITIONS** | | | | | | | | | |
| Household with density >2 per dormitory** | | -0.717 (0.056) | 0,033 | 2.325 | **0.020** | -0.164 (0.082) | -0.177 | -1.991 | **0.047** |
| Household with garbage collection** | | 0.096 (0.041) | -0,004 | -0.263 | 0.793 | -0.104 (0.187) | -0.038 | -0.558 | 0.577 |
| Household connected to the water supply** | | -0.009 (0.036) | -0,075 | -3.674 | **<0.001** | 0.158 (0.127) | 0.095 | 1.244 | 0.214 |
| Household connected to the sewer system** | | -0.795 (0.216) | -0,289 | -12.873 | **<0.001** | -0.390 (0.276) | -0.115 | -1.414 | 0.158 |
| **JOB CHARACTERISTICS** | | | | | | | | | |
| Commerce** | | -2.213 (0.116) | -0,281 | -19.109 | **<0.001** | -0.314 (0.207) | -0.086 | -1.515 | 0.131 |
| Informal workers** | | 0.245 (0.055) | 0,074 | 4.469 | **<0.001** | 0.136 (0.179) | 0.077 | 0.761 | 0.447 |
| **SOCIOECONOMIC AND INEQUALITIES CHARACTERISTICS** | | | | | | | | | |
| GINI Index** (ref = low)[a] | Medium | -0.775 (0.977) | -0,011 | -0.793 | 0.428 | -1.847 (1.475) | -0.084 | -1.252 | 0.211 |
| | High | 1.015 (1.163) | 0,015 | 0.873 | 0.383 | -3.645 (1.811) | -0.174 | -2.012 | **0.045** |
| Income per capita | | -0.049 (0.040) | -0,337 | -12.360 | **<0.001** | -0.015 (0.003) | -0.445 | -5.010 | **<0.001** |
| Informal urban settlements (%) | | -0.546 (0.117) | -0,056 | -4.665 | **<0.001** | -0.022 (0.049) | -0.023 | -0.437 | 0.662 |
| **HEALTH SERVICES ACCESS AND COVERAGE** | | | | | | | | | |
| Health coverage (ref = low)[a] | Medium | -2.052 (0.914) | -0,030 | -2.246 | **0.025** | 0.291 (2.041) | 0.008 | 0.143 | 0.887 |
| | High | 0.241 (0.996) | 0,003 | 0.242 | 0.809 | -1.638 (1.706) | -0.066 | -0.960 | 0.338 |

*increase every 10pp

a: variables classified according to tertiles

imported cases are likely to be higher in larger cities where health services quality and quantity are better as compared to smaller cities [36]. In addition, the incidence for 50/100,000 deaths was reported first in the Northeast region of the country. This pattern is confirmed by findings from another study that found that COVID-19 survival rates were lower in the North and Northeast regions [37]. Social, environmental and health characteristics could lead to this finding as in these regions access to and quality of health services and overall living conditions are lower as compared to other regions of the country [14, 18, 38]. Finally, we also found that age characteristics of the population was associated reaching 50 deaths/100,000 inhabitants. Cities

**Table 3. Factors associated to number of days since first case of COVID-19 in city until reach 1,000 cases/100,000 inhabitants (n = 4,848).**

| | | <100,000 | | | | ≥100,000 | | | |
|---|---|---|---|---|---|---|---|---|---|
| | | β (Std. error) | Std. Beta | t | p | β (Std. error) | Std. Beta | t | p |
| **GEOGRAPHIC REGION** | | | | | | | | | |
| Region (ref = Midwest) | North | 0.446 (4.495) | 0.004 | 0.099 | 0.921 | -12.400 (12.015) | -0.118 | -1.032 | 0.303 |
| | Northeast | -23.410 (4.716) | -0.107 | -4.964 | **<0.001** | -29.493 (13.090) | -0.208 | -2.253 | **0.025** |
| | Southeast | 34.077 (4.168) | 0.262 | 8.175 | **<0.001** | 3.855 (10.911) | 0.047 | 0.353 | 0.724 |
| | South | 30.931 (4.152) | 0.216 | 7.450 | **<0.001** | 29.782 (11.895) | 0.268 | 2.504 | **0.013** |
| Metropolitan (ref = no) | | 11.704 (3.384) | 0.054 | 3.459 | **0.001** | -14.601 (22.704) | -0.034 | -0.643 | 0.521 |
| Urban or rural (ref = urban) | | -13.061 (2.004) | -0.095 | -6.517 | **<0.001** | -2.698 (5.725) | -0.032 | -0.471 | 0.638 |
| **SOCIAL AND ENVIRONMENTAL CHARACTERISTICS** | | | | | | | | | |
| Urban population** (ref = low)[a] | Medium | -3.185 (2.193) | -0.026 | -1.453 | 0.146 | -1.367 (19.155) | -0.007 | -0.071 | 0.943 |
| | High | -2.235 (2.954) | -0.018 | -0.757 | 0.449 | 18.049 (19.205) | 0.106 | 0.940 | 0.348 |
| Population older than 60 years** (ref = low)[a] | Medium | 4.685 (2.297) | 0.037 | 2.040 | **0.041** | 13.375 (5.775) | 0.151 | 2.316 | **0.021** |
| | High | 1.347 (2.615) | 0.011 | 0.515 | 0.606 | 19.458 (9.229) | 0.128 | 2.108 | **0.036** |
| Indigenous population**[a] | | 2.016 (1.059) | 0.028 | 1.903 | 0.057 | -4.035 (5.013) | -0.048 | -0.805 | 0.421 |
| Black population** (ref = low)[a] | Medium | 3.955 (2.755) | 0.032 | 1.436 | 0.151 | 15.818 (6.981) | 0.181 | 2.266 | **0.024** |
| | High | 2.417 (3.434) | 0.019 | 0.704 | 0.482 | 24.327 (9.551) | 0.276 | 2.547 | **0.011** |
| Illiterate older than 25 years** (ref = low)[a] | Medium | -8.994 (2.692) | -0.072 | -3.341 | **0.001** | -23.358 (9.200) | -0.208 | -2.539 | **0.012** |
| | High | -20.246 (4.353) | -0.163 | -4.651 | **<0.001** | -52.532 (17.560) | -0.210 | -2.992 | **0.003** |
| City in extreme poverty (ref = no) | | 5.133 (2.059) | 0.040 | 2.492 | **0.013** | 26.610 (18.607) | 0.080 | 1.430 | 0.154 |
| **HOUSING CONDITIONS** | | | | | | | | | |
| Household with density >2 per dormitory** | | -0.756 (0.119) | -0.167 | -6.365 | **<0.001** | -0.323 (0.361) | -0.087 | -0.895 | 0.372 |
| Household with garbage collection** | | 0.401 (0.091) | 0.072 | 4.382 | **<0.001** | -0.679 (0.930) | -0.054 | -0.730 | 0.466 |
| Household connected to the water supply** | | -0.304 (0.078) | -0.074 | -3.902 | **<0.001** | -0.360 (0.562) | -0.052 | -0.640 | 0.522 |
| Household connected to the sewer system** | | 0.784 (0.461) | 0.041 | 1.700 | 0.089 | 1.963 (1.214) | 0.146 | 1.617 | 0.107 |
| **JOB CHARACTERISTICS** | | | | | | | | | |
| Commerce** | | 1.059 (0.246) | 0.074 | 4.297 | **<0.001** | 0.255 (0.912) | 0.018 | 0.280 | 0.780 |
| Informal workers** | | 0.016 (0.118) | 0.003 | 0.139 | 0.889 | -1.097 (0.793) | -0.156 | -1.385 | 0.167 |
| **SOCIOECONOMIC AND INEQUALITIES CHARACTERISTICS** | | | | | | | | | |
| GINI Index** (ref = low)[a] | Medium | 10.570 (2.093) | 0.084 | 5.049 | **<0.001** | 6.005 (6.479) | 0.069 | 0.927 | 0.355 |
| | High | 11.131 (2.497) | 0.088 | 4.459 | **<0.001** | 10.930 (7.942) | 0.132 | 1.376 | 0.170 |
| Income per capita | | -0.072 (0.008) | -0.275 | -8.604 | **<0.001** | -0.045 (0.013) | -0.338 | -3.430 | **0.001** |
| Informal urban settlements (%) | | -0.608 (0.246) | -0.034 | -2.468 | **0.014** | -0.375 (0.217) | -0.101 | -1.731 | 0.085 |
| **HEALTH SERVICES ACCESS AND COVERAGE** | | | | | | | | | |
| Health coverage (ref = low)[a] | Medium | -4.543 (1.951) | -0.037 | -2.329 | **0.020** | 14.816 (9.008) | 0.109 | 1.645 | 0.101 |
| | High | -5.034 (2.122) | -0.039 | -2.372 | **0.018** | -1.058 (7.554) | -0.011 | -0.140 | 0.889 |

*increase every 10pp

a: variables classified according to tertiles

with greater aging population were first impacted as compared to their counterparts. Older adults (>60 years) are among one of the main risk groups for COVID-19 [6] and are among the those reporting greater need of ICUs [17, 37, 39, 40]. In Europe, this age group had a major role in COVID-19 dissemination due the age composition and large number of long term-care facilities [41–43]. Overall, cities with greater social, economic and health disparities were first and most impacted by COVID-19 deaths (e.g., higher GINI index, lower literacy and income and higher proportion of indigenous populations). Similar findings have been reported in site specific studies in Brazil and in high-income countries confirming the role of

**Table 4. Factors associated to the number of days since first death by COVID-19 in city until reach 50 deaths/100.000 inhabitants (n = 2,991).**

| | | <100,000 | | | | ≥100,000 | | | |
|---|---|---|---|---|---|---|---|---|---|
| | | β (Std. error) | Std. Beta | t | p | β (Std. error) | Std. Beta | t | p |
| **GEOGRAPHIC REGION** | | | | | | | | | |
| Region (ref = Midwest) | North | 3.492 (6.905) | 0.023 | 0.506 | 0.613 | 9.051 (15.367) | 0.064 | 0.589 | 0.556 |
| | Northeast | -12.501 (6.879) | -0.053 | -1.817 | 0.069 | -13.391 (17.096) | -0.067 | -0.783 | 0.434 |
| | Southeast | 16.186 (6.170) | 0.106 | 2.623 | **0.009** | 17.580 (14.023) | 0.153 | 1.254 | 0.211 |
| | South | 24.356 (6.192) | 0.148 | 3.934 | **<0.001** | 25.639 (15.423) | 0.168 | 1.662 | 0.098 |
| Metropolitan (ref = no) | | 4.566 (5.261) | 0.018 | 0.868 | 0.386 | -31.489 (34.325) | -0.044 | -0.917 | 0.360 |
| Urban or rural (ref = urban) | | -1.242 (2.916) | -0.008 | -0.426 | 0.670 | -31.289 (7.593) | -0.263 | -4.121 | **<0.001** |
| **SOCIAL AND ENVIRONMENTAL CHARACTERISTICS** | | | | | | | | | |
| Urban population** (ref = low)[a] | Medium | 0.263 (3.460) | 0.002 | 0.076 | 0.939 | 30.909 (23.785) | 0.109 | 1.299 | 0.195 |
| | High | 7.282 (4.447) | 0.052 | 1.638 | 0.102 | 42.787 (24.063) | 0.180 | 1.778 | 0.076 |
| Population older than 60 years** (ref = low)[a] | Medium | -3.370 (3.448) | -0.023 | -0.977 | 0.328 | -0.174 (7.489) | -0.001 | -0.023 | 0.981 |
| | High | -13.833 (3.916) | -0.096 | -3.533 | **<0.001** | -20.592 (11.974) | -0.097 | -1.720 | 0.087 |
| Indigenous population**[a] | | 8.535 (1.631) | 0.101 | 5.234 | **<0.001** | 8.112 (6.647) | 0.069 | 1.220 | 0.223 |
| Black population** (ref = low)[a] | Medium | 2.648 (4.173) | 0.018 | 0.635 | 0.526 | -16.381 (9.263) | -0.136 | -1.768 | 0.078 |
| | High | -4.679 (5.269) | -0.031 | -0.888 | 0.375 | 6.936 (12.832) | 0.057 | 0.540 | 0.589 |
| Illiterate older than 25 years** (ref = low)[a] | Medium | -16.464 (4.051) | -0.114 | -4.064 | **<0.001** | -35.871 (11.481) | -0.230 | -3.124 | **0.002** |
| | High | -15.513 (6.836) | -0.105 | -2.269 | **0.023** | -29.502 (21.245) | -0.087 | -1.389 | 0.166 |
| City in extreme poverty (ref = no) | | 3.242 (3.226) | 0.021 | 1.005 | 0.315 | 10.882 (28.091) | 0.019 | 0.387 | 0.699 |
| **HOUSING CONDITIONS** | | | | | | | | | |
| Household with density >2 per dormitory** | | -0.157 (0.182) | -0.031 | -0.864 | 0.388 | -1.117 (0.486) | -0.217 | -2.298 | **0.022** |
| Household with garbage collection** | | -0.222 (0.160) | -0.030 | -1.387 | 0.165 | -0.405 (1.084) | -0.027 | -0.374 | 0.709 |
| Household connected to the water supply** | | -0.558 (0.134) | -0.104 | -4.165 | **<0.001** | -0.563 (0.745) | -0.063 | -0.756 | 0.451 |
| Household connected to the sewer system** | | 2.087 (0.679) | 0.096 | 3.075 | **0.002** | 0.641 (1.547) | 0.034 | 0.414 | 0.679 |
| **JOB CHARACTERISTICS** | | | | | | | | | |
| Commerce** | | 2.643 (0.374) | 0.159 | 7.059 | **<0.001** | 0.127 (1.178) | 0.006 | 0.108 | 0.914 |
| Informal workers** | | -0.408 (0.182) | -0.058 | -2.241 | **0.025** | -1.472 (1.016) | -0.151 | -1.450 | 0.148 |
| **SOCIOECONOMIC AND INEQUALITIES CHARACTERISTICS** | | | | | | | | | |
| GINI Index** (ref = low)[a] | Medium | 13.371 (3.159) | 0.092 | 4.233 | **<0.001** | -4.583 (8.321) | -0.038 | -0.551 | 0.582 |
| | High | 18.667 (3.853) | 0.126 | 4.845 | **<0.001** | -0.565 (10.203) | -0.005 | -0.055 | 0.956 |
| Income per capita | | -0.028 (0.013) | -0.092 | -2.192 | **0.028** | -0.011 (0.017) | -0.062 | -0.670 | 0.504 |
| Informal urban settlements (%) | | -1.299 (0.307) | -0.077 | -4.228 | **<0.001** | -1.342 (0.277) | -0.267 | -4.841 | **<0.001** |
| **HEALTH SERVICES ACCESS AND COVERAGE** | | | | | | | | | |
| Health coverage (ref = low)[a] | Medium | 1.897 (2.988) | 0.013 | 0.635 | 0.526 | -8.017 (11.522) | -0.043 | -0.696 | 0.487 |
| | High | -4.451 (3.191) | -0.031 | -1.395 | 0.163 | -7.042 (9.630) | -0.051 | -0.731 | 0.465 |

*increase every 10pp

a: variables classified according to tertiles

social determinants of health on COVID-19 mortality [13, 33, 44, 45]. It is important to point determinants could be different in Brazil in different periods, where the most exposed and vulnerable groups to the disease changed over time [43].

The ecological analysis of the first year of COVID-19 in Brazil highlighted important lessons for the ongoing and future pandemics. Pandemic preparedness should include city level surveillance allowing fast track cases and deaths to inform measure to mitigate spread of the disease (e.g., reducing crowding) and targeting those most vulnerable (e.g., elderly populations, those living in informal settings). Additionally, effort to deploy and enhance health

services in cities located in the regions most impacted by COVID-19 could greatly mitigate the impacts of future outbreaks and pandemics. In our results, cities with less than 100,000 inhabitants were associated with less days to reach, this could indicate a better health system capacity to detected COVID-19 cases, whereas probably other cities remained underreported. Places with better health coverage and preparedness, the COVID-19 burden were lower [46]. Finally, reducing social, economic and health disparities should be at the forefront of public health policies to address the social determinants of health. This ecological study findings should be interpreted with caution. Biases on case and deaths reporting are likely are no warranty, as surveillance systems quality vary by city size and location, especially in small cities. In addition, we included data from an array of sources which covered different time frames, most of those without a recent update (ranged from 2010 to 2021). When small cities are compared to other municipalities, their indicators are limited, which may affect regional inferences. Nevertheless, we strive to achieve the best distribution and comparability. Long term-care facilities, despite the increasing number in Brazil [47], were not included in our model due lack of proper information (distribution, quality, and surveillance during the COVID-19 pandemic) [48] and not culturally accepted yet by Brazilian population due social stigma related to poor quality of life and abandon [49]. Finally, the ecological nature of this study prevent any further inference on causality.

## Conclusion

Social and built environment characteristics and inequalities were associated with COVID-19 cases spread and mortality incidence in Brazilian cities. These characteristics have had greater impact in smaller cities as compared to larger cites. Future efforts to mitigate the ongoing and future pandemics should include reducing social, economic and health inequalities and enhancing public health systems in smaller cities.

## Supporting information

**S1 Table. Spearman correlation matrix for variables tested in this study for each dependent variable.**
(DOCX)

**S2 Table. Collinearity diagnostics for variables tested in this study for each dependent variable.**
(DOCX)

**S1 Fig. Advance of COVID-19 in Brazil after 1st case and 1st death every 3, 6, 9 and 12 months.**
(DOCX)

**S2 Fig. Populational distribution in Brazil.**
(DOCX)

## Author Contributions

**Conceptualization:** Alexandre Augusto de Paula da Silva, Rodrigo Siqueira Reis, Franciele Iachecen, Adriano Akira Ferreira Hino.

**Data curation:** Alexandre Augusto de Paula da Silva, Franciele Iachecen.

**Formal analysis:** Alexandre Augusto de Paula da Silva, Adriano Akira Ferreira Hino.

**Methodology:** Alexandre Augusto de Paula da Silva.

**Supervision:** Adriano Akira Ferreira Hino.

**Writing – original draft:** Alexandre Augusto de Paula da Silva, Adriano Akira Ferreira Hino.

**Writing – review & editing:** Rodrigo Siqueira Reis, Fábio Duarte, Cristina Pellegrino Baena, Adriano Akira Ferreira Hino.

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
