## [Decision Letter · Decision Letter 0]

13 Dec 2022

PONE-D-22-21698First year of COVID-19 in Brazil: Factors associated with the spread of COVID-19 in small and large citiesPLOS ONE

Dear Dr. Alexandre Augusto de Paula da Silva, 

Thank you for submitting your manuscript to PLOS ONE. After careful consideration, we feel that it has merit but does not fully meet PLOS ONE’s publication criteria as it currently stands. Therefore, we invite you to submit a revised version of the manuscript that addresses the points raised during the review process. There are major weaknesses in the methods that must be addressed in full before the article can be reconsidered for publication. We suggest that it could be re-presented with attention to times of occurrence of cases, deaths, etc. in different municipalities.

We look forward to receiving your revised manuscript.

Kind regards,

Fernanda Penido Matozinhos, Ph.D

Academic Editor

PLOS ONE

https://journals.plos.org/plosone/s/fileid=ba62/PLOSOne_formatting_sample_title_authors_affiliations.pdf.

“AAPS received PhD scholarship from Coordenação de Aperfeiçoamento de Pessoal de Nível Superior – Brasil (CAPES) – Finance Code 001.”

3. Thank you for stating the following in the Acknowledgments/Funding Section of your manuscript:

“The first author received PhD scholarship from Coordenação de Aperfeiçoamento de Pessoal de Nível Superior – Brasil (CAPES) – Finance Code 001.”

“AAPS received PhD scholarship from Coordenação de Aperfeiçoamento de Pessoal de Nível Superior – Brasil (CAPES) – Finance Code 001.”

4. We note that [Figure 1] in your submission contain [map/satellite] images which may be copyrighted. All PLOS content is published under the Creative Commons Attribution License (CC BY 4.0), which means that the manuscript, images, and Supporting Information files will be freely available online, and any third party is permitted to access, download, copy, distribute, and use these materials in any way, even commercially, with proper attribution. For these reasons, we cannot publish previously copyrighted maps or satellite images created using proprietary data, such as Google software (Google Maps, Street View, and Earth). For more information, see our copyright guidelines: http://journals.plos.org/plosone/s/licenses-and-copyright.

Natural Earth (public domain): http://www.naturalearthdata.com/.

Additional Editor Comments:

Dear Dr. Alexandre Augusto de Paula da Silva,

Thank you for submitting your manuscript to PLOS ONE. After careful consideration, we feel that it has merit but does not fully meet PLOS ONE’s publication criteria as it currently stands. Therefore, we invite you to submit a revised version of the manuscript that addresses the points raised during the review process. There are major weaknesses in the methods that must be addressed in full before the article can be reconsidered for publication. We suggest that it could be re-presented with attention to times of occurrence of cases, deaths, etc. in different municipalities.

Reviewers' comments:

Reviewer's Responses to Questions

**Comments to the Author**

1. Is the manuscript technically sound, and do the data support the conclusions?

Reviewer #1: Partly

Reviewer #2: Yes

2. Has the statistical analysis been performed appropriately and rigorously? 

Reviewer #1: No

Reviewer #2: No

3. Have the authors made all data underlying the findings in their manuscript fully available?

Reviewer #1: No

Reviewer #2: Yes

4. Is the manuscript presented in an intelligible fashion and written in standard English?

Reviewer #1: No

Reviewer #2: Yes

5. Review Comments to the Author

Reviewer #1: the article was well written. It has curious results.

However, it approaches the subject inappropriately. It looks like geographical determinism. It seems to me that aggregated data are not suitable for this kind of conclusion. Knowing the surveillance system in Brazil, it is very likely that smaller municipalities have more sensitive surveillance, which would make reading the data wrong.

Perhaps it can be re-presented in the form of a letter calling attention to these times of occurrence of cases, deaths, etc. in different municipalities.

Reviewer #2: This is an interesting article on Covid in Brazil but I do have some comments that will help the authors improve the paper.

First a note on the level of urbanization in Brazil. The authors state that "less than one out of ten" of the Brazilian population lives in cities greater than 100,000 people. No reference was provided. However, according to estimates based on the last census (2020), 123,022,512 people lived in the 326 cities with at least 100,000 people in Brazil on 1-7-2021 (IBGE 2021). That is 58%, not "less than one out of ten" of the Brazilian population. Moreover, Brazil's level of urbanization equalled 87.8% (World Bank), which is considered very high in world standards. In other words, urbanization and population density are much higher than the authors make it out to be. This is an important fact, because both factors were crucial in the initial spread of COVID, as has been shown elsewhere and both at the national level (for Europe, see e.g. Mogi and Spijker 2022) and in cities themselves (see e.g. Lopez-Gay et al 2022 in the case of Barcelona, Spain).

The authors state that "larger cities have better health care, transportation, and job access and opportunities". While this may the case, for COVID both urban transportation and job access were likely to be responsible for MORE not less cases. In Barcelona, Spain, for instance, most cases were observed during the first wave in the poorer, peripheral, neighbourhoods because many required to take public transport, particularly the metro, to go to work, and, due to the nature of their work (e.g. service industry) could not work from home. On the other hand, wealthier people could either take the car or work from home. Moreover, poorer people tend to live in more overcrowded and smaller dwellings. In addition, important urban areas have more "essential workers". As they are occupations that are over-represented by women and immigrants from low-income countries (at least in Spain), they also tend to live in poorer neighbourhoods and therefore more likely to commute. Overall, higher mortality rates from COVID-19 are associated with poorer neighbourhood conditions, including a scarcity of healthcare facilities. Lastly, the number of nursing and retirement homes has also been associated with a greater number of infections in the neighbourhood (see Lopez-Gay et al 2022 for references).

Lastly, there was no mention of nursing homes. Due to the age composition and poor health status of the residents and the confined and shared space they live in, infection and mortality rates were especially high during the first wave in the Barcelona Study that I earlier cited, and was partly responsible for the neighbourhood differences that were observed. While Brazil's population is less aged as that of high-income countries, it is ageing fast and as a result, the number of nursing homes are also increasing. However, their quality is often poor, especially those in non-capital cities (Fonseca de Oliveira et al 2017). Both therefore the number of nursing and other elderly homes and their quality in the different urban areas could therefore have had a repercussion on the infection and mortality rates.

Method: Why was the cut-off of 100,000 people used? Moreover, some of the smaller cities form part of the metropolitan area of one of Brazil´s mega cities like Rio or Sao Paolo, while others maybe much more isolated. The effect of being part of a Metropolitan area could be tested. Would it also be possible for the authors to repeat the test for the remaining (more) rural areas?

I also missed a theoretical justification for some of the covariates, especially after reading phrases such as "COVID-19 cases were 170 first reported in those cities with lower proportion of black residents", i.e. why would black residents have a higher or lower COVID rate? (please check Table 1 in Lopez-Gay et al where expected associations based on other studies is provided for the different variables that were tested).

A correlation matrix of the different independent variables and / or the VIF results should be included in the appendix.

Question: Why didn't the authors apply a generalized linear model such as a Quasi-Poisson regression that considers the total population as an offset?

Discussion: The authors should re-write the phrase "[the] arrival of imported cases is likely to be higher in larger cities where health services quality and quantity are better as compared to smaller cities leading to a possible bias in the reporting". One thing is who the virus arrived (e.g. by tourists, business trips), another is that cases may have been under-reported in areas where it took more time for the testing material to arrive and be used by the general population.

There was no discussion on the possible changes in the association between the tested factors and the level of covid during 2020. For instance, while the level of ageing may be important, in the paper by Mogi and Spijker (2022) the authors suggested that the (historical) level of economic development and (associated) social ties are likely to have led to the initial spread of the COVID-19 pandemic (in Europe) but that population density and cultural factors became more important for the subsequent diffusion of the virus. I therefore recommend that the authors perform some sensitivity analysis. Determinants are likely to be different from 0-10 cases per 100,000 than from 11-100 and from 101-1000. For prevention this would be useful information.

Lastly, given the ever-increasing number of studies on COVID, how do the results of this study compare to others? For instance, in related to the observed associations of the different variables?

References:

Brazilian Institute of Geography and Statistics (IBGE). ESTIMATIVAS DA POPULAÇÃO RESIDENTE NO BRASIL E UNIDADES DA FEDERAÇÃO COM DATA DE REFERÊNCIA EM 1º DE JULHO DE 2021 https://ftp.ibge.gov.br/Estimativas_de_Populacao/Estimativas_2021/estimativa_dou_2021.pdf

Fonseca de Oliveira, Wagner Ivan; et al (2017). Quality of Care In Nursing Homes In Brazil. Journal of the American Medical Directors Association, 18(7): 636-e13. doi:10.1016/j.jamda.2017.04.007

López-Gay A et al (2022). "Sociodemographic determinants of intraurban variations in COVID-19 incidence: the case of Barcelona." Journal of Epidemiology and Community Health 76:1-7. https://dx.doi.org/10.1136/jech-2020-216325

Mogi R and Spijker J. 2022 "The Influence of Social and Economic Ties to the Spread of Covid-19 in Europe." Journal of Population Research 39(4): 495-511. https://dx.doi.org/10.1007/s12546-021-09257-1.

United Nations Department of Economic and Social Affairs (UNDESA) (2018). World Urbanization Prospects: 2018 Revision. New York, UNDESA, Population Division. https://population.un.org/wup/

6. PLOS authors have the option to publish the peer review history of their article (what does this mean?). If published, this will include your full peer review and any attached files.

Reviewer #1: No

Reviewer #2: No

---

## [Author Response · Author response to Decision Letter 0]

5 Mar 2023

Dear Editor,

Regards the manuscript " First year of COVID-19 in Brazil: Factors associated with the spread of COVID-19 in small and large cities", we are sending this letter with responses of items suggested by reviewers. We thank the editor and reviewers for the time spent analyzing our manuscript and the suggestions and queries. The suggestions were properly addressed and answered, being highlighted on manuscript. Below this letter, the detailed responses for each comment sent from reviewers, with the respective indication on text, page, and line. We believe all requests have been answered.

Sincerely,

Authors

January 27th, 2023

---

## [Decision Letter · Decision Letter 1]

22 May 2023

PONE-D-22-21698R1First year of COVID-19 in Brazil: Factors associated with the spread of COVID-19 in small and large citiesPLOS ONE

Dear Dr. Alexandre Augusto de Paula da Silva,

Thank you for submitting your manuscript to PLOS ONE. After careful consideration, we feel that it has merit but does not fully meet PLOS ONE’s publication criteria as it currently stands. Therefore, we invite you to submit a revised version of the manuscript that addresses the points raised during the review process.

We look forward to receiving your revised manuscript.

Kind regards,

Fernanda Penido Matozinhos, Ph.D

Academic Editor

PLOS ONE

Additional Editor Comments:

Dear Alexandre Augusto de Paula da Silva,

Thank you for submitting your manuscript to PLOS ONE!

After careful consideration, we invite you to submit a revised version with substantial changes of the manuscript - that addresses the points raised during the review process – specially in the Method, Results and Discussion’ sections.

Kind regards,

Reviewers' comments:

Reviewer's Responses to Questions

**Comments to the Author**

1. If the authors have adequately addressed your comments raised in a previous round of review and you feel that this manuscript is now acceptable for publication, you may indicate that here to bypass the “Comments to the Author” section, enter your conflict of interest statement in the “Confidential to Editor” section, and submit your "Accept" recommendation.

Reviewer #2: (No Response)

Reviewer #3: (No Response)

2. Is the manuscript technically sound, and do the data support the conclusions?

Reviewer #2: Partly

Reviewer #3: Partly

3. Has the statistical analysis been performed appropriately and rigorously? 

Reviewer #2: No

Reviewer #3: Yes

4. Have the authors made all data underlying the findings in their manuscript fully available?

Reviewer #2: Yes

Reviewer #3: Yes

5. Is the manuscript presented in an intelligible fashion and written in standard English?

Reviewer #2: Yes

Reviewer #3: Yes

6. Review Comments to the Author

Reviewer #2: Dear authors,

I've reviewed the article again, but my main concern is still related to the way urbanisation is measured or dealt with. Brazil is a highly urbanised city. While only 1 in 10 cities have a population of 100,000 or more (as the authors have phrased it now), they pertain to 58% of Brazil's population, i.e. 9 out of 10 cities only contain 30% of the population as the remaining 12% is rural. This is an important fact as the majority of Brazilians actually have good excess to health care, jobs etc. if we consider that what the authors state (that larger cities have better health care, transportation, and job access and opportunities) is true. Yet, I'm still not convinced whether a city of 101,00 people actually has worse health care than one of 99,000 or 80,000 or even 50,000 if the latter is a regional centre and the former a poor suburb of a major city.

It was therefore good to see that the authors included the variables Metropolitan (yes/no) and an urban/rural covariate. However, I now wonder if there are too many variables that are highly correlated. For instance, between these two as well as the variables population density and urban population. I went through Table S1 and couldn't help noticing very similar associations between the dependent variables and the categories used for population density and urban population.

Likewise, regarding S2, the VIF for these in the models for cities >= 100,000 were particularly high for the regions (and incredibly high (VIF>100) for the indigenous population, suggesting very high correlation between variables. Often a VIF<5 is taken as threshold, although this threshold has also been debated (see https://doi.org/10.1007/S11135-006-9018-6 for more on this topic). I therefore think that some finetuning still needs to be done. Perhaps combining variables could be an option (e.g. crossing region with level of urbanisation). Given the relatively large number of covariates included in the study and the potential multicollinearity among them, another option is to run a lasso analysis to automatically identify the most relevant variables (see

Tibshirani R. Regression shrinkage and selection via the LASSO. Journal of the Royal Statistical Society: Series B 1996;58:267–88). In the context of generalised linear regression modelling and prediction, lasso performs both variable selection and regularisation to enhance prediction accuracy and interpretability of the statistical model.

Secondly, I wasn't satisfied by the answer regarding my point on LTC facilities. The population 60+ and household density are not necessarily proxies for LTC facilities. For the correlation between the population aged 70+ and beds in LTC facilities squared per 1000 people in 76 neighbourhoods in Barcelona during the first wave was only 0.27 and both variables were significant in the study (see my previous review for the reference). The number of nursing and other elderly home places per population in an urban area could therefore have had repercussions on the speed of infection and mortality rates. In another study it was shown that in 5 European countries about half of all COVID deaths occurred in care homes during April 2020: https://ltccovid.org/wp-content/uploads/2020/04/Mortality-associated-with-COVID-12-April-3.pdf As nursing homes are likely to be concentrated in specific (wealthier?) municipalities in Brazil, it is likely to have played a part in the initial spread of COVID, just like elsewhere. While Brazil's population is less aged as that of high-income countries, it is ageing fast and as a result, the number of nursing homes are also increasing (Fonseca de Oliveira et al 2017).

Reviewer #3: The manuscript focuses on topics related to the covid-19 pandemic and infectious conditions.

There are some points to adapt:

1. In the method, it is necessary to make clear what type of ecological design was applied: time series, multiple groups or mixed. This has implications for data analysis.

2. The outcome variables used do not have a semantic explanation of the indicators regarding their epidemiological usefulness. This has implications for the theoretical model of the study and for the conclusions.

3. In the results, there is no interpretation of the regression coefficients in the light of their sign and intensity, which weighs each of the independent variables on their importance and relationship with the outcomes.

4. Still in the results, it is necessary to present an interpretation on the differences between small and large cities regarding the independent variables.

5. In the discussion, it is necessary to raise some limitations of the databases used, mainly regarding their updating.

7. PLOS authors have the option to publish the peer review history of their article (what does this mean?). If published, this will include your full peer review and any attached files.

Reviewer #2: No

Reviewer #3: **Yes: **Johnnatas Mikael Lopes

---

## [Author Response · Author response to Decision Letter 1]

7 Jul 2023

We appreciate all the comments provided from the reviwers. All comments are presented in files attached. We hope all questions were properly fullfiled to improve the manuscript.

Best,

Authors

---

## [Decision Letter · Decision Letter 2]

18 Sep 2023

PONE-D-22-21698R2First year of COVID-19 in Brazil: Factors associated with the spread of COVID-19 in small and large citiesPLOS ONE

Dear Dr. Silva,

Thank you for submitting your manuscript to PLOS ONE. After careful consideration, we feel that it has merit but does not fully meet PLOS ONE’s publication criteria as it currently stands. Therefore, we invite you to submit a revised version of the manuscript that addresses the points raised during the review process.

The goal of this study is relevant and the manuscript offers a comprehensive perspective on the COVID-19 pandemic's transmission dynamics and its correlation with sociodemographic factors in Brazil.

We look forward to receiving your revised manuscript.

Kind regards,

Fernanda Penido Matozinhos, Ph.D

Academic Editor

PLOS ONE

Journal Requirements:

Additional Editor Comments:

Dear Alexandre Augusto de Paula da Silva,

Thank you for submitting your manuscript to PLOS ONE. After careful consideration, we feel that it has merit but does not fully meet PLOS ONE’s publication criteria as it currently stands. Therefore, we invite you to submit a revised version of the manuscript that addresses the points raised during the review process.

The goal of this study is relevant and the manuscript offers a comprehensive perspective on the COVID-19 pandemic's transmission dynamics and its correlation with sociodemographic factors in Brazil.

Kind regards,

Reviewers' comments:

Reviewer's Responses to Questions

**Comments to the Author**

1. If the authors have adequately addressed your comments raised in a previous round of review and you feel that this manuscript is now acceptable for publication, you may indicate that here to bypass the “Comments to the Author” section, enter your conflict of interest statement in the “Confidential to Editor” section, and submit your "Accept" recommendation.

Reviewer #4: All comments have been addressed

Reviewer #5: All comments have been addressed

2. Is the manuscript technically sound, and do the data support the conclusions?

Reviewer #4: Partly

Reviewer #5: Partly

3. Has the statistical analysis been performed appropriately and rigorously? 

Reviewer #4: Yes

Reviewer #5: Yes

4. Have the authors made all data underlying the findings in their manuscript fully available?

Reviewer #4: No

Reviewer #5: Yes

5. Is the manuscript presented in an intelligible fashion and written in standard English?

Reviewer #4: Yes

Reviewer #5: Yes

6. Review Comments to the Author

Reviewer #4: Review of Augusto de Paula da Silva et. al.

It is clear that the paper has been through prior cycles of review and I can see that the authors have tried to respond to the reviewer comments. Overall I think that it is a scientific contribution at a level appropriate for PloS One.

After reviewing, I have three main comments:

1) When considering regions in terms of things like 1,000 cases/ 100,000 inhabitants. I am concerned that there may be some bias depending on the number of inhabitants. Looking at Figure 1, many of the areas in the upper left of the map are very large and they reached the threshold very rapidly 1-50 days and when that occurs very large regions are affected. Part of this is that if you have a region with, for example 100 people, the very first case meets the threshold. This is a very different condition than a city of 10,000,000 where a 100,000 cases must be reached before the threshold is met. That city might be very small in terms of area. It is unclear how best to deal with this, perhaps another map colour coded by the number of people in the region. This would also help understand the areas that never reached thresholds for cases or deaths. As a minimum some discussion of the issue should appear more prominently.

2) While I may have missed it or not have access, I did not find the raw data. I think it is important that all the raw data be made available so others can check the analysis. I think this might be helpful for the authors as well. A careful PloS reviewer found a mistake in an analysis done in a paper I was co-author on. I was quite grateful. As presented here no one can check.

3) Similarly, it has been a few years since I last used SPSS (I switched to R because it is more powerful, free, and easy to provide scripts); however, having clarity on the model as done in SPPSS (perhaps a saved analysis) would help as well as some discussion of the model building process. The tables are great, but having the original specifications for the analysis makes it more transparent exactly how it was done. I seem to recall that some level of automation can be done in SPSS including all terms and interactions. Understanding the overall model building would improve the paper. Again, with clarity everything can be checked completely transparently. This allows interested readers to understand better.

Overall, it is a good contribution and deserves publication.

Reviewer #5: The manuscript presents data, results, and discussions of interest for an enhanced understanding of the transmission of COVID-19 among cities and regions of Brazil and its relationship with socio-demographic variables. It drew attention to the absence of any mention of the well-documented fact of rapid transmission among small cities in the southern region of Brazil, particularly concentrated among workers in the large meatpacking plants in this region. It is worth mentioning this issue, as its analysis can be found in various previously published articles. Given this exceptional situation that characterized the initial phase of the pandemic in the Southern Region, once it is included in this manuscript I am in favor of publishing it.

7. PLOS authors have the option to publish the peer review history of their article (what does this mean?). If published, this will include your full peer review and any attached files.

Reviewer #4: No

Reviewer #5: **Yes: **Helena Maria Scherlowski Leal David

---

## [Author Response · Author response to Decision Letter 2]

20 Oct 2023

Editorial Board

PLOS ONE

We are writing this letter with our responses to the reviewers comments our paper titled "First year of COVID-19 in Brazil: Factors associated with the spread of COVID-19 in small and large cities". We thank the editor and reviewers for the thoughtful and valuable suggestions and inquiries which have helped improving the quality of the manuscript. We are confident that we have adequately addressed and answered all the suggestions, which have been appropriately incorporated into revised document.

We are including detailed response to each of the reviewers' comments, indicating the corresponding page and line numbers in the text. We are confident that all requests and suggestions have been thoroughly addressed.

Sincerely,

Authors

---

## [Decision Letter · Decision Letter 3]

21 Dec 2023

PONE-D-22-21698R3First year of COVID-19 in Brazil: Factors associated with the spread of COVID-19 in small and large citiesPLOS ONE

Dear Dr. Alexandre Augusto de Paula da Silva,

Thank you for submitting your manuscript to PLOS ONE. After careful consideration, we feel that it has merit but does not fully meet PLOS ONE’s publication criteria as it currently stands. Therefore, we invite you to submit a revised version of the manuscript that addresses the points raised during the review process.

We look forward to receiving your revised manuscript.

Kind regards,

Fernanda Penido Matozinhos, Ph.D

Academic Editor

PLOS ONE

Journal Requirements:

Additional Editor Comments:

Dear Author,

After careful consideration, we feel that it has merit and we invite you to submit a revised version of the manuscript that addresses the points raised during the review process.

Kind regards,

Reviewers' comments:

Reviewer's Responses to Questions

**Comments to the Author**

1. If the authors have adequately addressed your comments raised in a previous round of review and you feel that this manuscript is now acceptable for publication, you may indicate that here to bypass the “Comments to the Author” section, enter your conflict of interest statement in the “Confidential to Editor” section, and submit your "Accept" recommendation.

Reviewer #4: All comments have been addressed

Reviewer #6: All comments have been addressed

2. Is the manuscript technically sound, and do the data support the conclusions?

Reviewer #4: Yes

Reviewer #6: Yes

3. Has the statistical analysis been performed appropriately and rigorously? 

Reviewer #4: Yes

Reviewer #6: Yes

4. Have the authors made all data underlying the findings in their manuscript fully available?

Reviewer #4: Yes

Reviewer #6: Yes

5. Is the manuscript presented in an intelligible fashion and written in standard English?

Reviewer #4: Yes

Reviewer #6: Yes

6. Review Comments to the Author

Reviewer #4: The reviewers have addressed my previous comments. I think the paper can be published. The analysis is valid and similar to other papers published previously but with a broader analysis of socioeconomic variables.

I do have a few comments, but since I did not raise these earlier, I do not think it appropriate to delay publication.

Comments:

In particular, the inclusion of the data set used is quite helpful.

Related to this, I would encourage the authors to consider the following.

1) The data is in SPSS “.sav” format which is a proprietary format. I was able to translate it into xlsx using this online tool: https://secure.ncounter.de/SpssConverter. I would encourage the authors as a minimum to convert it to xlsx format. However, a csv file would be better.

2) It was possible to find information about the enter method in SPSS, however, not being an SPSS user currently it is unclear how SPSS deals with the multiple comparisons problem. Briefly, we expect about 1 in 50 (p = 0.05) tests made on random data to show significance at the p < 0.05 level. The vast majority of significant effects appear robust (p < 0.001) and almost assuredly will hold up, but a few of the significant effects are more marginal (income per capita, high illiteracy, etc.) may be. Ideally, SPSS should do this by default but may not. This is worth checking.

3) Looking at the data set made me wonder about the white regions in the maps representing areas not reaching a particular threshold. In the data set, these appear as missing data. When I sort the data by population, the lowest population is 781 and the largest is 12,252,023 (quite a wide range).

At the same time, the largest regional (1500347) population not having a death had 27430 people; the largest region (2905206) not having 10 deaths was 50975; the largest with 50 deaths (2611101) was 349145; and the largest (3106200) not having 100 deaths was 2512070 people. All of these end up in the data set as missing data in these columns.

If I do a quick plot of (largest population, threshold for missing data), there is an approximately logarithmic relationship. With some effort, more points could be added, but this is an approximate minimum population threshold to observe a specific number of deaths. Figure 1 uses 50 deaths, but there are around 40% of regions that never got to 50 deaths. This makes me think that the populations sizes as defined by the COD_IBGE system are insufficient (too many small regions) to produce a good map with a 50 death threshold. It may be more useful to present 10 deaths (only 8% never reached this threshold). In addition, the smaller the region, the greater likelihood that it might never reach a particular threshold.

4) Finally, some of the effects seen may also reflect reporting variations rather than purely disease propagation. The analysis still holds but some wording to indicate this may be appropriate. For example: Region 1715507, population 1112, reported its 1st, 10th, and 50th death on the same day (7 October 2020). It is possible that ~5% of the population of the region died on a single day, however, there seems to be pattern of this in the data set. The following all reported 1st, 10th, and 50th death on the same day: Region: 5201207 on 17 July, 3555901 on 12 May, 4215695 on 25 November, … There a many more. That all these milestones were passed on the same day in 1 or two places is believable but I count 40+, all small regions, that show this.

Thank you again for providing the data set.

Reviewer #6: The study is an analysis of factors associated with the emergence of cases and deaths from Covid-19 in small and large cities in Brazil. It has an attractive analysis proposal with relevance for epidemiological research, especially for the study of pandemics

Title: suitable

Introduction: Raises justification and relevance of the study. However, it includes an excessive amount of text to raise points already well discussed in other articles or even available in official documents. It should be reduced and address a little more of the risk factors that were associated with the outcomes studied. The priority should be to present what is new about the article.

The method is adequately described.

Results: They are presented appropriately and meet the proposed objectives.

Discussion:

I suggest delving a little deeper into the hypotheses that could explain the results found. Even if the type of study does not allow a direct association of cause and effect, hypotheses that attempt to explain the outcomes listed are necessary. Not all outcomes received deserved attention in the discussion. For example, the reasons for the association of cases and deaths from Covid-19 in cities with greater health service coverage were not clear enough.

As a suggestion, some additional references can be consulted: 1-Guedes MBOG, de Assis SJC, Sanchis GJB, Araujo DN, Oliveira AGRDC, Lopes JM (2021) COVID-19 in Brazilian cities: Impact of social determinants, coverage and quality of primary health care. PLoS ONE 16(9): e0257347. https://doi.org/10.1371/journal.pone.0257347. 2- Costa de Assis SJ, Lopes JM, de Lima Filho BF, José Bouzas Sanchis G, Sousa Rodrigues Guedes T, Limeira Cavalcanti R, et al. (2021) Dissemination of COVID-19 in inland cities of Northeastern Brazil. PLoS ONE 16(7): e0253171. https://doi.org/10.1371/journal.pone.0253171).

Conclusion: it is adequate and follows what brings the objectives and results, without extrapolating.

References: they are mostly up to date and contain a number of journals with a good, considerable impact factor.

7. PLOS authors have the option to publish the peer review history of their article (what does this mean?). If published, this will include your full peer review and any attached files.

Reviewer #4: No

Reviewer #6: **Yes: **MARCELLO BARBOSA OTONI GONÇALVES GUEDES

---

## [Author Response · Author response to Decision Letter 3]

15 Jan 2024

Editorial Board

PLOS ONE

We are writing this letter with our responses to the reviewers comments our paper titled "First year of COVID-19 in Brazil: Factors associated with the spread of COVID-19 in small and large cities". We thank the editor and reviewers for the thoughtful and valuable suggestions and inquiries which have helped improving the quality of the manuscript. We are confident that we have adequately addressed and answered all the suggestions, which have been appropriately incorporated into revised document.

We are including detailed response to each of the reviewers' comments, indicating the corresponding page and line numbers in the text. We are confident that all requests and suggestions have been thoroughly addressed.

Sincerely,

Authors

---

## [Decision Letter · Decision Letter 4]

31 Jan 2024

First year of COVID-19 in Brazil: Factors associated with the spread of COVID-19 in small and large cities

PONE-D-22-21698R4

Dear Dr. Alexandre Augusto de Paula da Silva,

We’re pleased to inform you that your manuscript has been judged scientifically suitable for publication and will be formally accepted for publication once it meets all outstanding technical requirements.

Kind regards,

Fernanda Penido Matozinhos, Ph.D

Academic Editor

PLOS ONE

Additional Editor Comments (optional):

Dear Author,

Thank you for the opportunity to review this manuscript. I am grateful for the invitation.

After careful consideration, I feel the manuscript explores a very important topic. The questions were responded and modifications in the text made the manuscript come to a satisfying result.

Kind regards,

Fernanda Penido

Reviewers' comments:

Reviewer's Responses to Questions

**Comments to the Author**

1. If the authors have adequately addressed your comments raised in a previous round of review and you feel that this manuscript is now acceptable for publication, you may indicate that here to bypass the “Comments to the Author” section, enter your conflict of interest statement in the “Confidential to Editor” section, and submit your "Accept" recommendation.

Reviewer #4: All comments have been addressed

Reviewer #6: All comments have been addressed

2. Is the manuscript technically sound, and do the data support the conclusions?

Reviewer #4: Yes

Reviewer #6: Yes

3. Has the statistical analysis been performed appropriately and rigorously? 

Reviewer #4: Yes

Reviewer #6: Yes

4. Have the authors made all data underlying the findings in their manuscript fully available?

Reviewer #4: Yes

Reviewer #6: Yes

5. Is the manuscript presented in an intelligible fashion and written in standard English?

Reviewer #4: Yes

Reviewer #6: Yes

6. Review Comments to the Author

Reviewer #4: (No Response)

Reviewer #6: The manuscript includes acceptable methodological aspects and has undergone reinforcement in the discussion of its results and theoretical framework. After adjustments, the article is suitable for publication

7. PLOS authors have the option to publish the peer review history of their article (what does this mean?). If published, this will include your full peer review and any attached files.

Reviewer #4: No

Reviewer #6: No

---

## [Editor Report · Acceptance letter]

24 May 2024

PONE-D-22-21698R4 

PLOS ONE

Dear Dr. Silva, 

I'm pleased to inform you that your manuscript has been deemed suitable for publication in PLOS ONE. Congratulations! Your manuscript is now being handed over to our production team.

Kind regards, 

on behalf of

Dr. Fernanda Penido Matozinhos 

Academic Editor

PLOS ONE